# Associations Between Fundamental Movement Skills, Muscular Fitness, Self-Perception and Physical Activity in Primary School Students

**DOI:** 10.3390/jfmk9040272

**Published:** 2024-12-13

**Authors:** Andrew Sortwell, Rodrigo Ramirez-Campillo, Aron Murphy, Michael Newton, Gregory Hine, Ben Piggott

**Affiliations:** 1School of Health Sciences and Physiotherapy, University of Notre Dame Australia, 2 Mouat St, Fremantle, WA 6160, Australia; aron.murphy@nd.edu.au (A.M.); newton7269@outlook.com (M.N.); 2School of Education, University of Notre Dame Australia, 128-140 Broadway, Chippendale, NSW 2007, Australia; 3Research Centre in Sports, Health and Human Development, University of Beira Interior, 6201-001 Covilhã, Portugal; 4Exercise and Rehabilitation Sciences Institute, School of Physical Therapy, Faculty of Rehabilitation Sciences, Universidad Andres Bello, Santiago 7501015, Chile; ramirezcampillo@gmail.com; 5School of Education, University of Notre Dame Australia, 2 Mouat St, Fremantle, WA 6160, Australia; gregory.hine@nd.edu.au (G.H.); benjamin.piggott@nd.edu.au (B.P.)

**Keywords:** kinesiology, motor performance skills, self-concept, muscle function, neuromuscular performance, education

## Abstract

**Background/Objectives:** Positive self-perception, physical activity and fundamental movement skill (FMS) proficiency are important aspects of a child’s healthy development. The objective of this research was twofold: first, to explore associations between FMS, muscular fitness, self-perception and physical activity in school children; and second, to identify key predictors of FMS proficiency, athletic self-perception, physical activity levels in these participants and the differences between biological sexes. **Methods:** Primary school-aged children (n = 104; 53.85% female) from 8 to 10 years old (M = 9.04, SD = 0.69) engaged in two days of testing. All students were measured on FMS, standing long jump (SLJ), countermovement jump (CMJ), seated medicine ball chest throw and self-perception, followed by correlation and stepwise multiple linear regression analyses. **Results:** The findings revealed significant positive correlations between FMS proficiency, lower body muscular fitness (CMJ, SLJ), and athletic self-perception for the entire cohort, with varying results between male and female subgroups. Additionally, athletic self-perception showed strong relationships with other self-perception domains. Notably, forward stepwise regression analysis identified FMS proficiency as a significant predictor of physical activity levels, explaining 12.8% of the variance. SLJ, scholastic perception, and physical activity explained 45.5% of FMS variance for females. In males, FMS proficiency significantly predicted physical activity, accounting for 13.3% of the variance. SLJ and athletic competence self-perception explained 42.1% of FMS variance in males. **Conclusions:** The study indicates that FMS, self-perception, muscular fitness, and physical activity levels may mutually enhance each other, and that there is a need for biological sex-specific strategies to be considered in physical education programs.

## 1. Introduction

Physical activity is vital for children’s holistic development, contributing to cognitive, psychosocial and physical development [1,2]. Despite this, inadequate physical activity levels for children globally are widely recognized as a growing public health concern [3]. According to Guthold, Stevens, Riley and Bull [3], a pooled analysis of surveys with over 1.6 million adolescents showed that overall physical activity scores are also very low/poor worldwide. In addition, for children, females are less likely to engage in adequate levels of physical activity in comparison to males [4,5,6], due to factors such as a lack of opportunities for sport and active play for girls, cultural and societal norms, lower confidence in physical abilities, and limited access to female-specific sports programs or facilities [6,7]. This concerning issue of inadequate physical activity begins in childhood, as children who do not engage in sufficient physical activity during childhood are more likely to continue the same pattern of engagement in physical activity in adolescence, through their youth into adulthood and continue throughout the remainder of their lifespan [8,9]. Furthermore, a lack of physical activity during childhood and adolescence is closely associated with low health and skills-related fitness levels since physical activity provides the necessary stimulus for improved fitness [10,11].

Researchers have speculated that fundamental movement skills (FMS) proficiency is a determining factor in the worldwide prevalence of inadequate physical activity levels among children [12,13]. However, the paradox is that higher physical activity levels may lead to improved proficiency in FMS, as they have more opportunities to practice running, jumping, climbing, passing, throwing, and catching, which form the foundation for more complex physical activities [14]. The reasoning here is that as children engage more in physical activities, they increase their opportunities to practise and refine their non-sport specific FMS (i.e., running, jumping, throwing, and catching), thus becoming more proficient, providing a stronger foundation or starting point to develop complex movement skills (e.g., layup in basketball). In support of this contention, previous studies have shown a positive effect of interventions (i.e., after-school physical activity programs) that aim to increase time engaged in physical activity, resulting in improved movement competence [15,16,17]. To this end, the Australian Physical Education Curriculum for children in Grades 3 and 4 emphasizes the need for engagement in physical activities to refine movement skills in diverse contexts [18]. Children’s muscular fitness [19] and self-perception [20] also appear to be important determinants of FMS that warrant further investigation. Therefore, it is also entirely plausible and necessary to investigate a more complex construct beyond levels of physical activity, one that is associated with FMS proficiency, such as children’s muscular fitness [19,21,22] and self-perception [17,20].

A lack of muscular fitness, such as neuromuscular power, can limit a child’s motor ability to perform FMS effectively and furthermore, to then engage in a wide range of physical activities and sports with confidence and competence [19,23]. Cyclical bouts of physical activity can develop minimal muscular fitness levels; conversely, a reduction in physical activity can lead to a reduction in skeletal muscle size, strength and neuromuscular power [24]. Certain types of physical activities that involve skipping, hopping, jumping, and bodyweight exercises, delivered in a structured program (e.g., physical education classes, school fitness programs), have been demonstrated to be highly effective in improving both neuromuscular functioning and FMS [19,21]. Further, interventions stimulating the neuromuscular system, such as plyometrics, integrative neuromuscular training and other forms of resistance activities in school children, are efficacious in improving FMS proficiency and sport-specific skills compared to practice and skill instruction alone [25,26,27,28,29,30].

Enhancing muscular fitness in children and adolescents not only improves their ability to produce force that contributes to overall physical development [31,32] but also appears to enhance their athletic competence (i.e., perception of one’s capacity to excel in athletic pursuits and the demonstration of athletic prowess) [33]. Specifically, Lubans and Cliff [34] identified a correlation between muscular strength and physical self-perceptions, overall physical self-worth and global self-esteem in adolescents. These noted enhancements of self-perception, self-concept and self-esteem are also further supported by Collins, Booth, Duncan and Fawkner [21], who reported that muscular fitness-building activities can positively affect perceived physical strength, physical self-worth, and global self-worth in youth.

Athletic competence, as a specific domain of self-perception, has particular value in sports and physical activity contexts as it relates to one’s perception of their body’s movement capabilities, aspects of their sports ability and willingness to demonstrate ability [35]. Indeed, it does not constitute a reflective self-description or evaluation; rather, it represents a situation-specific self-assessment of perceived capability [36,37]. Previous studies have demonstrated that physical responses to exercise, as assessed by physical measures such as fitness testing, can influence physical self-efficacy through perceptions of what the body can achieve (e.g., sports competence, distance jumped, and muscular strength) [36,38] with consistent biological sex differences [39,40]. Hence, the assessment of functional qualities may contribute to the self-realization of one’s ability and comparison to others [41]. Furthermore, biological sex may also influence the general perceptions that children have of their ability to achieve personal success in games, physical activities, and sports and to demonstrate athletic prowess [42].

Since lifelong engagement in physical activity is critical to a healthier lifespan, investigating the key determinants of a strong foundation for ongoing participation in physical activity, such as FMS and athletic competence self-perception, is necessary. This study seeks to extend existing knowledge by exploring associations between FMS, muscular fitness, self-perception and physical activity and to identify the key predictors of fundamental movement skill proficiency, athletic competence, physical activity levels in school children and the differences between biological sexes.

## 2. Materials and Methods

### 2.1. Participants

This study is a secondary analysis utilising baseline data from 104 children (56 females and 48 males), 102 of whom fully participated in both baseline and post-tests of a previously published intervention study [43]. While the initial publication examined the effects of a 6-week intervention, the current observational study seeks to investigate associations between fundamental movement skills (FMS), muscular fitness, self-perception, and physical activity in primary school children, aiming also to identify predictors of athletic competence self-perception, FMS proficiency, and physical activity. Although both studies share participant data, they address distinct research questions and apply different statistical analyses.

The inclusion criteria were students aged 8 to 10 years, in Grades 3 or 4 and without current musculoskeletal impairments. After receiving University ethics approval (University of Notre Dame Australia, No. 2022-051S), students and their parent/s or guardian/s received a written explanation of all procedures in a research information sheet. In addition, both students, parents and guardians were offered the opportunity to attend a face-to-face afternoon and two online evening information sessions. At these sessions, the study was explained, and those present had an opportunity to ask the research team questions. Subsequently, students and their parents provided informed consent to participate in the study by signing the study consent form. Parents or guardians, along with their child, completed a physical activity readiness questionnaire (PARQ) to confirm that the child was physically healthy and met the inclusion criteria to safely participate in the study. All participants were informed that they could withdraw from the study at any time without penalty. The study was conducted according to the guidelines of the 2013 Declaration of Helsinki. For this study, the required sample size was calculated using G. Power 3.1.9.7 software before data collection. For regression analysis, a minimum sample size of 103 participants was computed with a sample size set at the 95% reliability level, standardized effect size at α = 0.05 level and theoretical power at 0.80.

### 2.2. Procedures

Prior to the research study testing data collection sessions, a full day was dedicated to providing familiarization sessions to students on the assessment protocols to minimize possible learning effects during the formal data collection. At the conclusion of the familiarization sessions, students were instructed to avoid engaging in strenuous physical activity or sports the day prior to testing. Testing was conducted over two days in an undercover protected area to eliminate the effects of environmental and weather conditions on performance. On Day One of testing, the following tests were conducted: anthropometric measurements, standing long jump (SLJ), and the Canadian Agility and Movement Skill Assessment (CAMSA). On Day Two, seated medicine ball chest throw, countermovement jump, and Self-Perception Profile for Children (SPPC) questionnaire were administered. For all testing, students wore their school sports uniform. All research assistants who administered the assessments were trained to conduct the tests. Students completed a warm-up activity consisting of jogging, running, lower-body callisthenics, and whole-body dynamic stretches prior to commencing the physical assessment tests. All students completed a brief familiarisation overview before performing each test described in the following sections.

#### 2.2.1. Anthropometric Measurements

On the first day of testing, anthropometric measurements were collected using an electronic scale (SECA 803, Hamburg, Germany), and mobile stadiometer (SECA 217, Hamburg, Germany), in accordance with standardized procedures from the International Society for the Advancement of Kinanthropometry guidelines by a trained research assistant. Body height (±0.1 cm) was measured using a stadiometer, with the participant standing barefoot. Body mass was measured to the nearest 0.1 kg using electronic scales, with the students standing barefoot, with feet together. Student height and weight measurements were used to calculate body mass index (BMI) using the following formula: weight [kg]/(height [m])^2^.

#### 2.2.2. Canadian Agility and Movement Skill Assessment Test

FMS proficiency was assessed using the CAMSA test, a valid and reliable tool for evaluating children’s FMS competence [44]. The test was administered following the protocol described by Longmuir et al. [45]. The CAMSA test requires students to complete a designated 20 m course that involves a series of sequenced movement skills as follows: (i) jumping with two feet in and out of three hoops on the ground; (ii) sliding from side to side over a 3 m distance; (iii) catching a squellet ball; (iv) throwing the squellet at a wall target 5 m away; (v) skipping for 5 m; (vi) hopping on one foot in and out of six hoops on the ground; and (vii) kicking a stationary soccer ball between two cones that are 5 m away and placed 2 m apart. A skill score is determined based on the CAMSA criterion-referenced assessment, and the total time taken to complete the movement course is recorded. Both the time and the skill are scored out of a maximum of 14 points each, and the overall raw score is calculated from the sum of the skill and time criteria scores, with a maximum of 28 points that can achieved. Following the prescribed protocol [44], students were given two demonstrations and two practice trials, followed by two timed and scored attempts. During practice trials, verbal cues were provided as specified in the protocol, while in test trials, cues were only given to remind children of the upcoming task. All overall scores were recorded, with the highest score used for analysis.

#### 2.2.3. Standing Long Jump Test

Student lower body muscular power was assessed using a valid and reliable SLJ test protocol [46]. The horizontal jump distance was measured using two parallel 10-metre fibreglass metric tapes placed 40 cm apart and taped to the floor using clear tape. The measuring tape commenced from the marked starting line, and the jump distance was measured at the nearest centimeter. Two research assistants measured and then recorded the distance. The students were given an explanation and demonstration of the jump, then an initial trial jump. After completion of the trial jump, the SLJ test was performed three times, separated by one minute of rest. The furthest jump distance result was used for further analysis.

#### 2.2.4. Seated Medicine Ball Chest Throw

On day two of testing, upper body muscular power was assessed using the valid and reliable seated medicine ball chest throw assessment [47]. The assessment was conducted according to the protocol described by Faigenbaum et al. [48]. The student sat with their back against a wall to perform the assessment, holding a 20 cm diameter 1 kg medicine ball powdered with magnesium carbonate. The student was instructed to lift the ball to their chest with their elbows touching the rear wall. On the direction from the research assistant, the student had to throw the ball up and out, without their back losing contact with the wall. Two ten-meter fibreglass metric tapes placed one meter apart, were affixed with clear tape to a black rubber floor mat to measure the distance from the wall to where the medicine ball landed. Students were given an explanation and demonstration of the seated medicine ball chest throw, which was then followed by an initial trial throw. Then the assessment was performed three times, separated by one minute of rest. The furthest throw distance was used for further analysis.

#### 2.2.5. Countermovement Jump

On day two of testing, the countermovement jump (CMJ), a valid and reliable assessment in children, was also used to assess lower-body muscular power [46]. Firstly, the student’s maximum reach height was measured using a Vertec (Yardstick, Swift Performance Equipment, Australia) followed by performance of the CMJ, assessed according to the previously described protocol [49]. Students had one initial practice attempt, followed by three subsequent recorded attempts. Two research assistants monitored and recorded the CMJ jump height. The overall CMJ distance was determined and calculated by subtracting the standing reach height from the jump height. The highest maximum jump height was used for subsequent analysis.

#### 2.2.6. Athletic, Scholastic, Social Competence, and Global Self-Worth

Student self-perception was assessed using the Self-Perception Profile for Children (SPPC) questionnaire [35], which is a reliable and validated psychometric measure in children [50]. The SPPC questionnaire included the assessment of four self-perception sub-domains: athletic, scholastic and social competence self-perception, and global self-worth. Each subdomain had six questions, each being answered on a four-point scale, with higher scores being a more positive self-perception. The students’ teacher was present in the classroom when the research assistant administered the SPPC questionnaire to the students, following the protocol described by Harter [35].

#### 2.2.7. Children’s Leisure Activity Study Survey

The parental proxy questionnaire of the Children’s Leisure Activity Study Survey (CLASS) was used to assess student level of engagement in moderate-to-vigorous physical activity (MVPA). It has demonstrated acceptable reliability for both frequency and duration of physical activity, with an intra-class correlation coefficient (ICC) of 0.69 and duration (ICC = 0.74) [51]. The CLASS includes a checklist of 30 physical activities for children, categorized into 18 moderate-intensity activities and 12 vigorous-intensity activities. Parents were asked to indicate whether their child participates in each activity during a typical school week and weekend, and if yes, to report the frequency and duration of participation. Total weekly time in hours of physical activity was calculated.

### 2.3. Statistical Analyses

Descriptive data were calculated for all variables and presented as group mean values ± SD (Table 1). The normality of the data was verified (*p* > 0.05) using the Shapiro–Wilk test. Descriptive variables for both biological sexes were compared using an unpaired t-test. The reliability of each test involving multiple trials was evaluated by calculating the ICC [52]. Associations of variables were assessed using Pearson product–moment correlation coefficient. Associations were reported by their correlation coefficient *r*, and level of significance. Values of *r* ≥ 0.10 and <0.30 indicate a small, *r* ≥ 0.30 and <0.50 a medium, and *r* ≥ 0.50 a large correlation [53]. In addition, multiple regression analyses were used to determine the most robust correlations and predictors of FMS proficiency, athletic competence self-perception and physical activity levels for the following groupings: entire cohort, males and females. Based on the results of the t-tests revealing a significant difference between males and females for six dependent variables (see Table 2), the current analyses involved a whole cohort and separate regression analyses for males and females [54]. A forward-entry stepwise multiple linear regression model (entry probability = 0.05; removal probability = 0.10) was constructed to obtain the variables to be used in the final model. Adjusted R-squared (*R*^2^) values indicated variations (%) in parameters affecting an outcome variable. The variables for the regression analysis were finalized after analyzing the tolerance test and variance inflation factor scores to assess for possible multicollinearity issues [55]. The tolerance of predictor variables were to be greater than 0.1 and variance inflation factor less than 5, for there to be no multicollinearity [56]. The Durbin–Watson statistic test was performed to detect the presence of autocorrelation, with an acceptable range assumed to be between 1.50 and 2.50 [57]. All analyses were performed using Statistical Package for Social Sciences for Windows (SPSS, Version 28.2, IBM Company, Armonk, NY, USA).

## 3. Results

Participants (*n* = 104, M = 9.04, SD = 0.69 years) were primary school children from 3rd and 4th Grade. Mean and standard deviations for student characteristics and physical activity levels are shown in Table 1. The results revealed only a significant difference in height between the biological sexes.

Table 2 presents descriptive data of the completed assessments by children in the study and a comparison between male and female performance. The results show differences between the biological sexes in seated medicine ball chest throw (*p* < 0.001), SLJ (*p* = 0.002), self-perception sub-domain scores that included athletic competence (*p* = 0.019), global self-worth (*p* = 0.023), scholastic competence (*p* = 0.018) and social competence (*p* = 0.006). The ICC for CAMSA and muscular fitness measures ranged from 0.825 to 0.98 (*p* < 0.001).

The relationships between the FMS, muscular fitness variables and self-perception domains for the entire cohort, males and females, are presented in Table 3. There was a medium correlation between FMS level of proficiency and athletic perception (*r* (104) = 0.486, *p* < 0.01) for the entire cohort. In the males and females, there were also medium correlations (*r* (48) = 0.464, *p* < 0.01) and (*r* (56) = 0.488, *p* < 0.01), respectively. There was a small correlation for the entire cohort between FMS and social self-perception (*r* (104) = 0.198, *p* < 0.05), with no correlation among the male and female groups. No correlation was observed between FMS and the other domains of self-perception in the entire cohort, and among the male and female groups. There was a small negative relationship between BMI and FMS (*r* (104) = −0.198, *p* < 0.05) for the entire cohort; however, in the male and female grouping, it was not significant (*p* > 0.05).

Significant positive correlations were detected between FMS and muscular fitness variables of CMJ and SLJ in the entire cohort and male and female groupings (Table 3). Respective *r* values ranged from 0.40 to 0.62 (*p* < 0.01). No significant correlation was observed between the female group’s FMS and muscular fitness variable seated medicine ball chest throw (*p* > 0.05). However, there were significant correlations between FMS and seated medicine ball chest throw in the entire cohort and male group, with respective *r* values ranging from 0.41 to 0.51 (*p* < 0.01).

There were significant positive correlations between athletic self-perception and muscular fitness variables, CMJ (*r* (104) = 0.254, *p* < 0.01), seated medicine ball chest throw (*r* (104) = 0.354, *p* < 0.01), SLJ (*r* (104) = 0.417, *p* < 0.01) for the entire cohort. A moderate positive correlation was also detected between athletic self-perception and CMJ (*r* (56) = 0.322, *p* < 0.05) in the female group, however, in contrast, the male group showed no significant correlation (*p* > 0.05) between these variables. A moderate correlation was observed between athletic self-perception and seated medicine ball chest throw (*r* (48) = 0.358, *p* < 0.05) in males; however, in contrast, the female group showed no significant correlation (*p* > 0.05). There was a large significant correlation between athletic self-perception and SLJ (*r* (56) = 0.535, *p* < 0.01) in the female group; however, there was no significant correlation (*p* > 0.05) in the male group.

Significant positive relationships were observed between athletic self-perception and other sub-domains of self-perception (global self-worth, social and scholastic self-perception) for the entire cohort and the male group, with correlations ranging from 0.243 to 0.389. However, in the female group, there was only a statistically significant correlation detected between athletic competence and the self-perception subdomain of social self-perception (*r* (56) = 0.296, *p* < 0.05). A small significant correlation was detected between athletic self-perception and physical activity level in the entire cohort (*r* (104) = 0.215, *p* < 0.05), but was absent in the male and female groupings.

There were significant relationships observed between physical activity and CAMSA, SLJ, athletic and social competence self-perception across the entire cohort with correlations ranging from 0.215 to 0.370 (see Table 3). In the male and female groupings, CAMSA and SLJ were statistically correlated to physical activity, ranging from 0.317 to 0.32, respectively. In the male grouping, social competence self-perception was also correlated to physical activity levels (*r* (48) = 0.298, *p* < 0.05) (see Table 3).

Forward stepwise multiple linear regression for the outcome variable fundamental movement skill for the entire cohort is presented in Table 4. Regression analysis was conducted with FMS as the outcome variable, which indicated that the following predictors explained variance: SLJ, self-perception of athletic competence, and physical activity level. The tolerance values ranged from 0.781 to 0.902. In addition, the variance inflation factor values ranged from 1.109 to 1.280. Thus, there were no multicollinearity issues present in this analysis. The Durbin–Watson test produced a score of 2.235, showing a score near to zero autocorrelation. The full model for the entire cohort in relation to predictors of FMS accounted for 44.1% of the variance in CAMSA (adjusted *R*^2^ = 0.441, *F*(1, 100) = 5.630, *p* = 0.02), with the statistically significant predictors being the SLJ (B = 0.079, *p* < 0.001, 95% CI [0.049, 0.109]); athletic self-perception (B = 0.263, *p* = 0.002, 95% CI [0.103, 0.424]); physical activity level (B = 0.126, *p* = 0.02, 95% CI [0.021, 0.232]). The predictive model shows that for each unit (cm) increase in standing long jump score, the CAMSA score would increase by 0.079 units, and for each unit increase in athletic self-perception score would result in an increase in 0.263 unit increase in CAMSA score. Additionally, each unit (hour) increase in physical activity level would result in a 0.126 unit increase in CAMSA score.

Standardized beta coefficients indicated that SLJ (β = 0.435, *p* < 0.001) had a larger impact on CAMSA scores compared to athletic competence self-perception (β = 0.265, *p* < 0.01), physical activity levels (β = 0.184, *p* = 0.02), suggesting that lower body muscular power is more important for enhancing FMS.

Forward stepwise multiple linear regression for the outcome variable athletic competence self-perception for the entire cohort is present in Table 4. Regression analysis with athletic competence as the outcome variable indicated that the following predictors explained variance: CAMSA, self-perception of social competence, and seated medicine ball chest throw. The tolerance values ranged from 0.80 to 0.829. In addition, the variance inflation factor values ranged from 1.041 to 1.251. As a result, no multicollinearity issues were present in this analysis. The Durbin–Watson test yielded a score of 2.159, suggesting minimal autocorrelation. The full model for the entire cohort in relation to predictors of athletic competence showed that the predictors explained 33.1% of the variance (adjusted *R*^2^ = 0.331, *F*(1, 100) = 4.693, *p* = 0.033), with the statistically significant predictors being CAMSA (B = 0.350, *p* < 0.001, 95% CI [0.170, 0.531]); self-perception of social competence (B = 0.307, *p* < 0.001, 95% CI [0.140, 0.475]); and seated medicine ball chest throw (B = 0.016, *p* = 0.033, 95% CI [0.001, 0.030]). The predictive model shows that for each unit increase in CAMSA, athletic competence self-perception would increase by 0.35 units, each unit increase in social competence would increase athletic competence by 0.307 units and lastly, each unit (cm) increase in medicine ball chest throw would result in a 0.016 unit increase in athletic competence score. Standardized beta coefficients indicated that CAMSA (β = 0.348, *p* < 0.001) had a larger impact on athletic competence self-perception compared to social competence self-perception (β = 0.299, *p* < 0.001), and medicine ball chest throw (β = 0.191, *p* < 0.05), suggesting that level of FMS proficiency is more important for a greater level of athletic competence self-perception.

Forward stepwise multiple linear regression analysis with physical activity levels as the outcome variable indicated that only CAMSA was a predictor. The tolerance values ranged from 0.640 to 0.989 for all potential predictors. In addition, the variance inflation factor values ranged from 1.011 to 1.562. Thus, no multicollinearity issues were present in this analysis. The Durbin–Watson test revealed a score of 2.181, suggesting a level of autocorrelation near zero. The full model for the entire cohort in relation to predictors of physical activity levels indicated that the predictors explained 12.8% of the variance (adjusted *R*^2^ = 0.128, *F* (1, 102) = 16.185, *p* < 0.05), with the statistically significant predictor being CAMSA (β = 0.538, *p* < 0.001, 95% CI [0.273, 0.804]). According to the predictive model, for each unit increase in CAMSA, physical activity levels would increase by 0.538 h per week.

For the female group, regression analysis conducted with FMS as the outcome variable indicated that the following predictors explained variance: SLJ, level of physical activity and scholastic competence self-perception (see Table 5). The tolerance values ranged from 0.858 to 0.955. In addition, the variance inflation factor values ranged from 1.047 to 1.166. Hence, no multicollinearity issues were present in this analysis. The Durbin–Watson test produced a score of 2.433, denoting a score near zero autocorrelation. The full model for the female grouping in relation to predictors of FMS explained 45.5% of the variance in CAMSA (adjusted *R*^2^ = 0.455, *F*(1, 52) = 5.315, *p* = 0.025), with the statistically significant predictors being the SLJ (B = 0.101, *p* < 0.001, 95% CI [0.062, 0.141]); Scholastic competence self-perception (B = 0.222, *p* = 0.011, 95% CI [0.053, 0.390]); physical activity level (B = 0.148, *p* = 0.025, 95% CI [0.019, 0.276]). The predictive model indicates that for each unit (cm) increase in SLJ score, CAMSA score would increase by 0.101 units, and for each unit increase in scholastic self-perception score would result in an increase in 0.222 unit increase in CAMSA score, and for each unit (hour) per week increase in physical activity level would result in a 0.148 unit increase in CAMSA score. Standardized beta coefficients indicated that SLJ (β = 0.540, *p* < 0.001) had a stronger impact on CAMSA scores compared to scholastic competence self-perception (β = 0.269, *p* < 0.011) and physical activity levels (β = 0.248, *p* = 0.025), emphasizing the greater importance of lower body muscular power for enhancing FMS in females.

For athletic competence self-perception as the outcome variable in females, regression analysis indicated that the only predictor of variance is the SLJ. The tolerance values for the potential predictors ranged from 0.555 to 1.00. In addition, the variance inflation factor values ranged from 1.042 to 1.801. Therefore, there were no multicollinearity issues in this analysis. The Durbin–Watson test resulted in a score of 1.849, representing a score close to zero autocorrelation. In the full model for the female grouping in relation to predictors of athletic competence, the prediction explained 27.3% of the variance (adjusted *R*^2^ = 0.273, *F*(1, 54) = 21.642, *p* < 0.001), with the statistically significant predictor being SLJ (B = 0.111, *p* < 0.001, 95% CI [0.063, 0.159]). The predictive model shows that for each unit (cm) increase in SLJ, athletic competence would increase by 0.111 units.

For regression analysis conducted with the female grouping and physical activity levels as the outcome variable, the results indicated that the following predictors explained variance: CAMSA and scholastic competence self-perception. The tolerance values for the potential predictor variables ranged from 0.597 to 0.999. In addition, the variance inflation factor values ranged from 1.001 to 1.676. Thus, no multicollinearity issues were detected in this analysis. The Durbin–Watson test produced a score of 2.161, suggesting levels of autocorrelation that are near zero. In the full model for the female grouping in relation to predictors of physical activity levels, 18.8% of the variance in CAMSA (adjusted *R*^2^ = 0.188, *F*(1, 53) = 5.592, *p* = 0.022), with the statistically significant predictors being the CAMSA (B = 0.724, *p* < 0.001, 95% CI [0.305, 1.143]) and scholastic competence self-perception (B = −0.408, *p* = 0.022, 95% CI [−0.754, −0.062]). According to the predictive model, for each unit increase in CAMSA score, physical activity levels would increase by 0.724 h per week, and each unit increase in scholastic competence self-perception score would result in a 0.408 h per week decrease in physical activity levels. Standardized beta coefficients indicated that CAMSA (β = 0.432, *p* < 0.001) had a larger impact on physical activity levels compared to scholastic competence self-perception (β = −0.295, *p* = 0.022), suggesting that the level of FMS proficiency is more important than scholastic competence self-perception for increasing physical activity levels.

For the male group, regression analysis conducted with FMS as the outcome variable indicated that the following predictors explained variance: SLJ and athletic competence (see Table 6). The tolerance values for the potential predictor variables ranged from 0.457 to 1.00. In addition, the variance inflation factor values ranged from 1.000 to 2.191. Hence, no multicollinearity issues were detected in this analysis. The Durbin–Watson test produced a score of 1.938, representing levels of autocorrelation near zero. The full model for the male grouping in relation to predictors of FMS explained 42.1% of the variance in CAMSA (adjusted *R*^2^ = 0.421, *F*(1, 45) = 9.434, *p* = 0.004), with the statistically significant predictors being the SLJ (B = 0.093, *p* < 0.001, 95% CI [0.050, 0.137]); athletic self-perception (B = 0.391, *p* = 0.001, 95% CI [0.135, 0.648]). The predictive model shows that for each unit (cm) increase in SLJ score, the CAMSA score would increase by 0.093 units. The model also predicts that for each unit increase in athletic competence self-perception score, CAMSA score would increase by 0.391 units. Standardized beta coefficients indicated that SLJ (β = 0.493, *p* < 0.001) had a larger impact on CAMSA scores compared to athletic competence self-perception (β = 0.350, *p* < 0.01), emphasizing the greater importance of lower body muscular power for enhanced FMS in males.

Regression analysis conducted for the grouping males with athletic competence self-perception as the outcome variable indicated that the following predictors explained variance: CAMSA, scholastic competence self-perception and social competence self-perception. The tolerance values ranged from 0.937 to 0.987. In addition, the variance inflation factor values ranged from 1.013 to 1.067. As a result, there were no multicollinearity issues present in this analysis. The Durbin–Watson test revealed a score of 2.011, suggesting minimal autocorrelation. The full model for the male grouping in relation to predictors of FMS explained 37.5% of the variance in CAMSA (adjusted *R*^2^ = 0.375, *F*(1, 44) = 5.303, *p* = 0.026), with the statistically significant predictors being the CAMSA (B = 0.395, *p* < 0.001, 95% CI [0.186, 0.605]); scholastic competence self-perception (B = 0.337, *p* = 0.016, 95% CI [0.066, 0.609]); and social competence (B = 0.280, *p* = 0.026, 95% CI [0.035, 0.526]); the predictive model indicates that for each unit increase in CAMSA score, athletic competence self-perception would increase by 0.395 units, and for each unit increase in scholastic self-perception score would result in 0.337 unit increase in athletic competence. The model also predicted that for every unit increase in social competence, there would be a 0.280 unit increase in athletic competence. Standardized beta coefficients indicated that CAMSA (β = 0.422, *p* < 0.001) had a larger impact on athletic competence self-perception, compared to scholastic competence self-perception (β = 0.297, *p* = 0.016), and social competence self-perception (β = 0.274, *p* = 0.026) suggesting that level of FMS proficiency is more important than scholastic and social competence self-perception.

A regression analysis was conducted for the male group, with physical activity levels as the outcome variable, and it was indicated that only CAMSA was a predictor of variance. The tolerance values for the potential predictor variables ranged from 0.671 to 1.000. In addition, the variance inflation factor values ranged from 1.011 to 1.491. Thus, no multicollinearity issues were present in this analysis. The Durbin–Watson test indicated a score of 2.161, representing a score close to zero autocorrelation. In the full model for the male grouping in relation to predictors of physical activity levels, the predictors explained 13.3% of the variance (adjusted *R*^2^ = 0.133, *F*(1, 46) = 8.202, *p* = 0.006), with the statistically significant predictors being CAMSA (B = 0.501, *p* = 0.006, 95% CI [0.149, 0.853]). According to the predictive model, for each unit increase in CAMSA, physical activity levels would increase by 0.501 h per week.

## 4. Discussion

Both attainment of FMS proficiency in childhood and engagement in physical activity are core priorities of the health and physical education curriculum [58]. The development of FMS proficiency in the early years of primary school education is seen as a precursor to the learning and mature performance of more complex skills required for participation in sports [14]. Previous research has reported that children with poor FMS were identified as being at risk of low levels of physical activity, lower levels of muscular fitness and emotional and social challenges [59,60]. Furthermore, in childhood, positive self-perception of movement skill competence has been suggested to be important for developing and contributing to further engagement in physical activity later in adolescence [61]. The current study investigated the relationships between FMS, physical activity levels, muscular fitness variables, various domains of self-perception and the key predictors of fundamental movement skill proficiency, athletic competence, physical activity levels in school children, and the differences between biological sexes.

The results of the current study indicate medium to large significant associations between all measures of muscular fitness (i.e., CMJ, seated medicine ball chest throw and SLJ) and FMS in the entire cohort. The results are in accordance with the conceptual model developed by Lämmle et al. [62], which suggests performance skills proficiency is strongly associated with muscular power or explosive strength, which aligns with the findings from systematic reviews investigating the correlates of gross motor competence [63,64]. This finding suggests that improvements in muscular fitness are likely to be associated with positive changes in FMS proficiency. Furthermore, the findings broadly support the work of other experimental design studies involving interventions such as plyometrics or integrative neuromuscular training that improve both muscular power, but also consequently enhance FMS proficiency, compared to physical education lessons alone [25,27,28].

The significant association between FMS and muscular fitness for the males in this study mirrored that of the entire cohort. In the females, there were significant associations between the muscular fitness measures of CMJ and SLJ with FMS, however, no significant correlation was found between FMS and seated medicine ball chest throw. The biological sex difference indicates potential insights into how specific muscular fitness attributes (i.e., upper body muscular power) relate to the overall performance of FMS within the context of a dynamic criterion-referenced based and performance test of FMS proficiency. It could be argued that greater levels of upper-body muscular power improve neuromuscular function, such as proprioception and postural control, which then contributes to the performance of FMS [65]. As a result, young male children may rely more on motor abilities such as upper body muscular power to perform movement skills effectively. Considering these findings and the significant differences in muscular power between male and female children in this study (see Table 2), the potential interference of different baselines cannot be ruled out.

It is unsurprising that there was a small negative relationship between BMI and FMS for the entire cohort, indicating that higher BMI might be slightly associated with lower FMS proficiency. An explanation for this finding maybe due to a greater BMI, reducing power to weight ratio, making it more difficult to perform FMS and maintain movement control, especially those movements that rely on explosive strength or muscular power such as hop, skip, jump, throw, kick and sprint. Since the CAMSA test is a dynamical criterion-referenced and performance test used to assess both technique and time to complete a series of movement skills, higher levels of neuromuscular muscular power serve as an advantage considering the movements skills (skip, hop, jump, throw, kick) required to be performed. Contrary to expectations, when analyzing the data separately for males and females, this relationship between BMI and FMS was not statistically significant (*p* > 0.05), suggesting that the observed trend in the overall sample does not hold consistently within the individual biological sexes. This finding highlights the complexity of the relationship between BMI and FMS and the need for further research to understand the underlying factors.

The relationships between athletic self-perception and muscular fitness variables were noteworthy. Athletic self-perception showed significant positive relationships with CMJ, seated medicine ball chest throw, and SLJ in the entire cohort, with correlations ranging from 0.254 to 0.417 (*p* < 0.01). Notably, biological sex differences were observed: in females, athletic self-perception correlated moderately with CMJ and SLJ but not with seated medicine ball chest throw. Conversely, seated medicine ball chest throw showed a significant correlation in males, but CMJ and SL did not. These differences highlight the varying contributions of specific muscular fitness components to athletic self-perception across biological sexes. Furthermore, the findings indicate that different aspects of muscular fitness might be more closely associated with athletic self-perception in males versus females.

Athletic self-perception was also significantly correlated with other subdomains of self-perception, such as global self-worth, social competence self-perception, and scholastic competence self-perception, particularly in males. These findings suggest that males who perceive themselves as competent in athletics tend to have higher overall self-esteem and perceive themselves positively in social and academic contexts. In contrast, among females, significant correlations were found only between athletic competence self-perception and social competence self-perception, indicating that athletic competence self-perception might primarily influence their social self-view. This study also identified a significant but small correlation between athletic competence self-perception and physical activity levels in the entire cohort (*r* = 0.215, *p* < 0.05), though this was not evident within the biological subgroups. This finding suggests that while athletic competence self-perception can motivate physical activity, other factors may also play a substantial role.

The results indicated a medium positive correlation between FMS proficiency and athletic self-perception across the entire cohort (*r* = 0.486, *p* < 0.01), and similarly, medium correlations were observed within both males (*r* = 0.464, *p* < 0.01) and females (*r* = 0.488, *p* < 0.01). These findings suggest that children more proficient in FMS tend to have higher perceptions of their athletic abilities. The results are also in accordance with Piek et al. [66], who reported that gross motor ability affects perceived athletic competence in children with developmental coordination disorder. Indeed, the findings support the notion that FMS competence may play a critical role in developing a child’s athletic self-concept [61,67]. According to Peers et al. [68] it could be argued that higher levels of FMS proficiency enhance confidence in one’s athletic capabilities, which may, in turn, promote greater engagement in physical activities. However, the findings in this study showed that there was a small association between athletic competence and physical activity levels across the whole sample, but no significant association in the individual sexes. It is likely, therefore, that when data were grouped by both biological sexes, the sample size within each subgroup may have become too small to detect significant associations and aggregating data into larger groups increase statistical power to reveal associations that are not detectable within smaller subgroups.

The current study observed a small significant correlation between FMS and social self-perception for the entire cohort (*r* = 0.198, *p* < 0.05), but interestingly, no significant correlation was found within the biological sex male and female groupings. This observation suggests that while there is a general association between FMS and social self-perception, it may not be as robust or consistent across different biological sex groupings. Furthermore, this disparity could also imply that the influence of FMS on social self-perception might be mediated by other factors not examined in this study, such as peer interactions or social support systems.

The regression analyses provided insights into predictors of FMS, athletic competence self-perception and physical activity. For the entire cohort, SLJ, athletic self-perception, and physical activity levels emerged as significant predictors of FMS, explaining 44.1% of the variance. This finding underscores the importance of these factors in enhancing FMS proficiency. Of the three predictors in the model, SLJ was the largest predictor of FMS proficiency, suggesting that the development of this skill and associated performance is important for primary school student’s FMS development. Furthermore, the findings further justify and affirm the effects of interventions, such as plyometrics and integrative neuromuscular training, within primary schools for enhancing neuromuscular power, and likely improving FMS [19].

Athletic competence was the second largest predictor of FMS proficiency for the whole cohort and the grouping males. This relationship may partly be explained by Harter [69] competence motivation theory, suggesting that individuals who perceive themselves as highly competent are more likely to persist in certain activities and practice compared to those with lower perceived competence. Furthermore, individuals in achievement-oriented situations tend to choose activities that enhance their sense of competence [70]. Our findings also affirm previous and more recent research that has observed the positive relationship between motor competence and perceived athletic competence [71,72,73]. One anticipated result of the analysis was that physical activity levels were also a predictor of FMS proficiency. Physical activity is necessary for motor development as it provides a natural process required for acquiring, refining and maintaining motor skills and motor abilities (i.e., muscular power, strength). Providing children in schools with opportunities during school breaks to engage in diverse physical activities, especially activities that are also shown to improve motor skills (i.e., running, skipping, hopping) and motor abilities (i.e., neuromuscular power), are likely to be more advantageous.

Biological sex-specific analyses revealed different predictors for FMS proficiency. For females, SLJ, physical activity level, and scholastic competence self-perception were significant predictors of FMS. For males, SLJ and athletic self-perception were key predictors of FMS. One unexpected outcome finding was that scholastic competence self-perception was a predictor in females. It is difficult to explain this result, but it may be related to females with a higher scholastic competence self-perception possessing better cognitive skills [74], such as attention, memory, and problem-solving, which can aid in learning and refining FMS. For instance, these children might be better at understanding and following instructions during sports and physical education classes, leading to accelerated learning and development of FMS. Overall, the findings suggest that interventions aimed at improving FMS and athletic competence self-perception may need to be differentiated and tailored differently for males and females to address these unique predictors effectively.

For athletic competence as the criterion variable in the whole cohort, significant predictors included FMS proficiency, self-perception of social competence, and seated medicine ball chest throw, explaining 33.1% of the variance. This highlights the multifaceted nature of athletic competence, which is influenced not only by physical attributes but also by social perceptions. The results are in accordance with previous research by Barnett, Morgan, van Beurden and Beard [20] that showed developing movement skills such as object control skills can foster the development of a higher self-perception of athletics competence in children. Biological sex-specific analyses revealed different predictors for athletic competence. For females, only SLJ predicted athletic competence self-perception, whereas for males, FMS, social competence and scholastic competence self-perception were significant predictors for athletic competence self-perception. Even though speculative, but aligned with Harter [69] competence motivation theory, it is quite plausible that the child knowingness of their ability to perform FMS or movements similar to SLJ and seated medicine ball chest throw, where their performance is easily compared to others, are more aware of their motor skills and abilities to participate with greater success in sport, hence higher athletic competence self-perception. An unanticipated result is social competence self-perception being a predictor of self-perceived athletic competence. It seems possible that successful engagement in physical activities contributes to social status and peer acceptance, which consequently influences self-efficacy in the athletic domain [75].

FMS proficiency was the only predictor of physical activity levels for the whole cohort and the grouped males and the largest predictor for the grouped females. These results are expected and align with the recent research by Garbeloto et al. [76] examining the relationship between FMS proficiency and physical activity levels in 6–10-year-olds. However, an additional finding in this study, is that the regression model for physical activity levels in females showed scholastic self-perception to have an inverse effect on physical activity levels in females. The negative unstandardized beta coefficient for scholastic self-perception (B = −0.408, *p* < 0.05) indicates that individuals with higher scholastic self-perception tend to engage in lower levels of physical activity. This could be interpreted as students who perceive themselves as academically competent might prioritize academic work [77,78], or predominantly choose cognitive engaging activities over physical activities, leading to reduced time participating in physical activities. The positive unstandardized beta coefficient for CAMSA (B = 0.724) suggests that females with higher CAMSA scores are more likely to engage in higher levels of physical activity. The lack of correlation between scholastic self-perception and CAMSA scores means there is no mediation effect. Instead, both variables independently influence physical activity levels. These independent effects highlight the importance of considering multiple dimensions of students’ lives when looking at predictors of physical activity. Interventions aimed at increasing physical activity might need to address these different dimensions separately, ensuring that improvements in one area (e.g., scholastic self-perception) do not inadvertently reduce physical activity levels.

### 4.1. Strengths and Limitations

It is important in any scientific study to acknowledge the strengths and limitations. A strength of this study was the selection of the tools used to measure FMS (CAMSA), muscular fitness (SLJ test, seated medicine ball chest throw, CMJ), and SPPC. All these assessments were shown to be valid and reliable measures that have all been used extensively in previous research. A further strength of this study is the statistical analysis, which included stepwise regressions to provide biological sex comparison over in addition to total cohort data. However, a potential limitation is the overall sample size. A larger sample would be advantageous; however, it is challenging to undertake research with participants in schools. Another potential limitation is the absence of accelerometers to measure physical activity levels in conjunction with the CLASS survey for a more accurate measure. A third limitation is that the correlation and regression analyses do no provide causal relationships (inherent to cross-sectional studies) between the studied variables, however, it does provide some insight into the relationships between variables, helping to understand how changes in one variable might be associated with changes in another. The fourth limitation is that Australian children with high levels of engagement in physical activity were recruited, limiting the extrapolation of current findings to other groups. Hence, we can only reasonably generalize the findings to Australian students who have a relatively high level of engagement in physical activity.

### 4.2. Implications and Future Research

This study underscores the intricate connections between physical proficiency and self-perception, emphasizing the need for comprehensive physical education programs that foster skill development and positive self-perception in primary school students. Such results have several practical implications for educators and coaches: (i) the strong associations between FMS proficiency and athletic and social self-perceptions highlight the importance of developing FMS in children; (ii) programs designed to enhance FMS could contribute to improved self-perceptions and overall physical fitness; (iii) the biological sex-specific differences observed suggest that tailored approaches might be necessary to address the unique needs of males and females. Future research should explore the underlying reasons for the biological sex differences and investigate additional factors that could influence the relationships between FMS, muscular fitness, and self-perception. Furthermore, future research should explore longitudinal impacts and potential interventions tailored to biological sex-specific needs to promote holistic physical and psychological development.

## 5. Conclusions

Muscular fitness, self-perception, athletic self-concept, and FMS are significantly related variables in primary school students aged 8–10 years. The relationships may be modulated by children’s biological sex, suggesting the need for specific strategies in physical education programs. Enhancing our understanding of these relationships can better support children’s physical and psychological development, potentially promoting a healthier and more active lifestyle. While further exploration of these relationships among children is warranted, children should have access to PE curriculum opportunities that develop a comprehensive range of FMS and muscular fitness to facilitate confident and competent students, allowing increased participation in diverse physical activities and sports, thus promoting a healthy, active lifestyle.

## Figures and Tables

**Table 1 jfmk-09-00272-t001:** Characteristics of the students.

Characteristics	Entire Cohort (*n* = 104)	Males (*n* = 48)	Females (*n* = 56)	*p*-Value
Biological sex (% female/male)	55.6/44.4			
Age (years)	9.04 ± 0.69	9.08 ± 0.71	9.00 ± 0.66	0.537
Height (cm)	138.12 ± 6.51	139.54 ± 6.26	136.91 ± 6.53	0.039 *
Weight (kg)	34.11 ± 7.47	35.24 ± 7.62	33.14 ± 7.28	0.155
BMI (kg/m^2^)	17.81 ± 3.24	18.02 ± 3.58	17.64 ± 2.93	0.550
Physical activity level (h/wk^−1^)	11.39 ± 5.68	11.31 ± 5.60	11.46 ± 5.80	0.895

Note: Data reported as mean ± SD or percentage; *: *p* < 0.05 between sex; cm: centimeters; kg: kilograms; BMI: body mass index; h/wk^−1^: hours per week.

**Table 2 jfmk-09-00272-t002:** Descriptive statistics.

Variable	Entire Cohort (*n* = 104)	Males (*n* = 48)	Females (*n* = 56)	*p*-Value
**Fundamental movement skills**				
CAMSA (score)	19.20 ± 3.90	19.75 ± 4.34	18.74 ± 3.46	0.190
**Muscular fitness**				
CMJ (cm)	27.73 ± 5.58	28.54 ± 6.33	27.04 ± 4.81	0.172
SMBCT (cm)	276.96 ± 47.75	295.96 ± 46.96	260.68 ± 42.53	<0.001 *
SLJ (cm)	134.56 ± 21.57	141.67 ± 22.96	128.46 ± 18.42	0.002 **
**Self-perception subdomains**				
Athletic CSP (score)	18.58 ± 3.94	19.55 ± 3.89	17.75 ± 3.82	0.019 *
Global self-worth (score)	19.28 ± 3.29	20.13 ± 3.31	18.66 ± 3.15	0.023 *
Scholastic CSP (score)	16.81 ± 3.94	17.79 ± 3.42	15.96 ± 4.19	0.018 *
Social CSP (score)	17.18 ± 3.83	18.30 ± 3.81	16.23 ± 0.42	0.006 **

Note: Data are reported as mean ± SD. Abbreviations: cm: centimeters; CAMSA: Canadian agility movement skill assessment (as a measure of fundamental movement skills); CMJ: countermovement jump; SMBCT: seated medicine ball chest throw; SLJ: standing long jump; CSP: competence self-perception; *, **: difference between males and females at *p* < 0.05 and *p* < 0.01, respectively.

**Table 3 jfmk-09-00272-t003:** Correlation between FMS, muscular fitness, sub-domains of self-perception, and basic characteristics (total *n* = 104, males *n* = 48, females *n* = 56).

		1 ^£^	2	3	4	5	6	7	8	9
**2. CMJ**	Total	0.415 ^b^								
	Male	0.410 ^b^								
	Female	0.396 ^b^								
**3. SMBCT**	Total	0.413 ^b^	0.422 ^b^							
	Male	0.511 ^b^	0.472 ^b^							
	Female	0.261	0.323 *							
**4. SLJ**	Total	0.600 ^b^	0.709 ^b^	0.556 ^b^						
	Male	0.574 ^b^	0.737 ^b^	0.517 ^b^						
	Female	0.620 ^b^	0.667 ^b^	0.484 ^b^						
**5. Athletic CSP**	Total	0.486 ^b^	0.254 ^b^	0.354 ^b^	0.417 ^b^					
	Male	0.464 ^b^	0.155	0.358 *	0.230					
	Female	0.488 ^b^	0.322 *	0.240	0.535 ^b^					
**6. GSW**	Total	0.105	0.003	0.061	−0.013	0.243 *				
	Male	0.125	0.156	0.200	0.013	0.389 ^b^				
	Female	0.026	−0.245	−0.246	−0.201	0.032				
**7. Scholastic CSP**	Total	0.131	−0.061	0.052	0.024	0.267 ^b^	0.554 ^b^			
	Male	−0.021	−0.105	−0.124	−0.113	0.350 *	0.518 ^b^			
	Female	0.219	−0.092	0.030	0.000	0.139	0.546 ^b^			
**8. Social CSP**	Total	0.198 *	0.097	0.062	0.188	0.380 ^b^	0.299 ^b^	0.285 ^b^		
	Male	0.102	0.002	−0.016	0.008	0.387 ^b^	0.216 ^b^	0.227 ^b^		
	Female	0.250	0.138	−0.067	0.234	0.296 *	0.289 *	0.247		
**9. BMI**	Total	−0.198 *	0.270 ^b^	0.244 *	0.273 ^b^	−0.009	0.138	0.210 *	0.037	
	Male	−0.238	−0.335 *	0.195	−0.356 *	0.073	0.299 *	0.147	0.225	
	Female	−0.168	−0.207	0.292 *	−0.243	−0.126	−0.580	0.256	0.201	
**10. Physical activity**	Total	0.370 ^b^	0.152	0.125	0.297 ^b^	0.215 *	0.019	−0.033	0.255 *	−0.013
	Male	0.389 *	0.133	0.241	0.317 *	0.261	0.049	0.221	0.298 *	0.020
	Female	0.367 ^b^	0.184	0.047	0.320 *	0.193	0.000	−0.200	0.185	−0.173

Note: * *p* < 0.05; ^b^
*p* < 0.01; ^£^: Canadian agility movement skill assessment (as a measure of fundamental movement skills); CMJ: countermovement jump; SMBCT: seated medicine ball chest throw; SLJ: standing long jump; CSP: competence self-perception; GSW: global self-worth; BMI: body mass index.

**Table 4 jfmk-09-00272-t004:** Stepwise multiple regression analysis for the whole sample (*n* = 104).

Variables	Unstandardized Beta (B)	Standardized Beta (β)	Confidence Interval		Adjusted *R*^2^
			Lower Bound	Upper Bound	
**CAMSA**					0.441 *
SLJ	0.079 ***	0.435	0.049	0.109	
Athletic CSP	0.263 **	0.265	0.103	0.424	
PA	0.126 *	0.184	0.021	0.232	
**Athletic CSP**					0.331 *
CAMSA	0.350 ***	0.348	0.170	0.531	
Social CSP	0.307 ***	0.299	0.140	0.475	
SMBCT	0.016 *	0.191	0.001	0.030	
**PA**					0.128 *
CAMSA	0.538 ***	0.370	0.273	0.804	

Note: Significant correlations at * *p* < 0.05, ** *p* < 0.01, *** *p* < 0.001. CAMSA: Canadian agility movement skill assessment (as a measure of fundamental movement skills); SLJ: standing long jump; CSP: competence self-perception; PA: physical activity; SMBCT: seated medicine ball chest throw.

**Table 5 jfmk-09-00272-t005:** Stepwise multiple regression analysis explaining variance in females (*n* = 56).

Variables	Unstandardized Beta (B)	Standardized Beta (β)	Confidence Interval		Adjusted *R*^2^
			Lower Bound	Upper Bound	
**CAMSA**					0.455 *
SLJ	0.101 ***	0.540	0.062	0.141	
Scholastic CSP	0.222 *	0.269	0.053	0.390	
PA	0.148 *	0.248	0.019	0.276	
**Athletic CSP**					0.273 ***
SLJ	0.111 ***	0.535	0.063	0.159	
**PA**					0.188 *
CAMSA	0.724 ***	0.432	0.305	1.143	
Scholastic CSP	−0.408 *	−0.295	−0.754	−0.062	

Note: Significant correlations at * *p* < 0.05, *** *p* < 0.001. CAMSA: Canadian agility movement skill assessment (as a measure of fundamental movement skills); SLJ: standing long jump; CSP: competence self-perception; PA: physical activity.

**Table 6 jfmk-09-00272-t006:** Stepwise multiple regression analysis explaining variance in males (*n* = 48).

Variables	Unstandardized Beta (B)	Standardized Beta (β)	Confidence interval		Adjusted *R*^2^
			Lower Bound	Upper Bound	
**CAMSA**					0.421 **
SLJ	0.093 ***	0.493	0.050	0.137	
Athletic CSP	0.391 **	0.350	0.135	0.648	
**Athletic CSP**					0.375 *
CAMSA	0.395 ***	0.422	0.186	0.605	
Scholastic CSP	0.337 *	0.297	0.066	0.609	
Social CSP	0.280 *	0.274	0.035	0.526	
**PA**					0.133 **
CAMSA	0.501 **	0.389	0.149	0.853	

Note: Significant correlations at * *p* < 0.05, ** *p* < 0.01, *** *p* < 0.001. CAMSA: Canadian agility movement skill assessment (as a measure of fundamental movement skills); CSP: competence self-perception; SLJ: standing long jump; PA: physical activity.

## Data Availability

The data that supports the findings of this study are available from the corresponding author upon request.

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
