# Peer review of "Associations Between Fundamental Movement Skills, Muscular Fitness, Self-Perception and Physical Activity in Primary School Students"

_jfmk, 2024, doi:10.3390/jfmk9040272_

Round 1
Reviewer 1 Report
Comments and Suggestions for Authors
Basic reporting
Dear authors, the manuscript is generally well-written and easy to read; a slight spell-check is required. I have just some concerns that the authors must address.
Abstract
You state that the principal aim of your research was to explore associations between FMS, muscular fitness, self-perception and physical activity in school children. Why physical activity is not mentioned in the title?
Introduction
The literature on the subject is sufficiently well summarised. However, it could be useful to add some information about:
- lines 68–70: muscular fitness is discussed as a determinant of FMS proficiency, but its role may not be as universally influential as implied. FMS development is also significantly influenced by cognitive, social, and environmental factors, which are not explored in detail.
- sex differences in physical activity levels and self-perception topic lacks clarity on how biological sex specifically influences these aspects, you could elaborate on this topic.
- Social, cultural, and environmental factors (e.g., access to facilities, parental influence, socioeconomic status) might impact physical activity levels and FMS proficiency, these aspects should be considered and discussed.
Methods
- Tests requiring high physical exertion (e.g., SLJ, CMJ) might influence the subsequent tests (e.g., SPPC questionnaire on self-perception). There’s a potential for fatigue or altered performance due to prior tests. Was the order of tests randomized or standardized to avoid carryover effects?
- The PAR-Q was completed by parents/guardians, but there is no mention of whether students were directly observed or medically screened for conditions not disclosed by parents. Similarly, parental proxy reporting of children's physical activity levels (via the CLASS survey) is prone to recall and social desirability bias, which could affect accuracy. This potential bias should at least state as limitations.
Validity of the findings
- Differences in predictors of FMS and self-perception based on biological sex are discussed but, what was your possible explanation of why these differences exist? Without exploring sociocultural, developmental, or biological explanations, these findings remain descriptive rather than explanatory.
- I could be wrong but: the negative relationship between BMI and FMS for the entire cohort is explained by reduced power-to-weight ratios. However, you note that this relationship is non-significant within male and female subgroups. This could raise the question of whether the observed trend in the whole cohort is statistically reliable or simply a product of aggregation bias.
Author Response
Dear Reviewer,
Thank you for your valuable suggestions and feedback, which we have carefully read and taken into consideration, along with the other reviewer's comments, in revising the manuscript.
Kindest regards

Reviewer 2 Report
Comments and Suggestions for Authors
This article presents a topic of high interest and is very relevant given the study population.
Introduction
It is necessary to expand the argumentation, qualifying which are, according to the bibliography, more important with respect to the age to which the work refers (especially in the FMS).
It is necessary to make some reference to the educational environment and the link between this information and the type of practice given in this educational cycle, since Physical Education is compulsory at this age.
Methods
The starting level of each child should be clarified further, given that the difference in physical condition may affect the final results.
It should be indicated whether the person who performs the anthropometric test is qualified to do so. The reason for this selection of tests could be justified to determine if it is a protocol used previously or an original idea.
It would be desirable to include an image of the procedure to provide the reader with a more accurate idea of the context.
Discussion
The relationship between self-perception and muscle fitness is very interesting; it should be noted that the effort perception scales are not fully validated below 12 years of age, which could be related to this data.
Author Response
Dear Reviewer,
Thank you for your valuable suggestions and feedback, which we have carefully considered, along with the other reviewer’s comments, in revising the manuscript.
Kindest regards

Reviewer 3 Report
Comments and Suggestions for Authors
Firstly, I appreciate the opportunity to review this paper. I will present a series of considerations that, in my opinion, could improve and clarify both the research objectives and the results and discussion sections of the paper.
In the introduction, the first paragraph (lines 41–51) states something obvious and widely known within the scientific community: the insufficient levels of physical activity and the fact that physical inactivity is associated with poor health outcomes.
Every introduction should begin by addressing what is known about the research topic but should avoid stating the obvious (this is not a journalistic article). Therefore, I suggest the authors focus on the specific topic (see the research objective) and clearly state what is known about the correlation of the variables analyzed. In summary, this first paragraph should either be removed or summarized in a single sentence.
In my opinion, the introduction should focus on the content described in the paragraph (lines 83–92) and expand on the scientific evidence (few authors are cited) regarding muscular factors and subjects’ self-perception across different factors.
The authors state that the research objective is to expand existing knowledge by exploring the associations between FMS, muscular fitness, self-perception, and physical activity, and to identify the main predictors of fundamental motor skill competence, athletic competence, physical activity levels in schoolchildren, and differences by biological sex. Therefore, the introduction should analyze the current scientific knowledge regarding each of the elements included in the objective.
I suggest that the authors rewrite the introduction, citing the most important studies on the association between FMS, muscular fitness, self-perception, and physical activity in a structured manner. Then, they should cite the most relevant studies on predictors of competence in FMS, athletic competence, and physical activity levels in schoolchildren.
After this, I suggest including a paragraph explaining why their study is important. The introduction should conclude with the objective (the objective, although very ambitious, is well written).
It is worth noting that scientific articles must adhere to the principle of economy in writing (i.e., it is unnecessary to repeat what has already been stated). Specifically, in section 2.1. Participants, the authors restate the objective. Why is this repeated when it is already clearly presented at the end of the introduction?
The authors indicate in line 124 an inclusion criterion, but they do not specify how it was applied. What data were used to exclude subjects? This should be clearly stated.
All information related to informed consent (line 128) should be included in the “Procedure” section. Additionally, considerations about the sample size should be clearly explained. Based on the information provided, the population is unclear. Was it randomized? If so, this should be specified.
In section 2.2.1. Did the authors use only the described formula to calculate BMI? Was there any adjustment based on the children's ages? Are the authors aware of the biases associated with applying this formula at these ages? Please explain this.
In section 2.3. Statistical Analysis, the authors should describe only the tools used for statistical estimations, nothing more. For example, references to results should not appear, as in lines 250–251: “Based on the results of t-tests that revealed significant differences between males and females for six dependent variables (see Table 2),”.
I completely disagree with the authors’ statement: “Values of r ≥ 0.10 and < 0.30 indicate a small correlation, r ≥ 0.30 and < 0.50 a medium correlation, and r ≥ 0.50 a large correlation.”
The reason is as follows: with a Pearson correlation coefficient of 0.1, the variables share only 1% of the variance; with r=0.3, 9%; and with r=0.5, the shared variance is 25%. These are residual values. For r=0.5, the variables share only 25% of the variance, which is insufficient to establish correlations, especially when referring to any kind of prediction. For the authors' study, a minimum correlation of 0.7 should be established. This corresponds to a shared variance of 49%, meaning less than half of the variance is shared. A good correlation would require shared variance above 60%.
I also suggest the authors indicate the p-value for the normality tests performed with the Shapiro–Wilk test.
The authors report in the results that there was a high correlation between FMS levels and athletic perception (r=0.486). The shared variance between the two variables is 23.61%, meaning that if plotted on a regression line, significant dispersion would be evident.
Additionally, the R² in Table 2 (regressions) explains very little of the variance, resulting in low predictive power. The authors should address this in the discussion.
Once the authors address these concerns, I will be happy to review the paper again.
Author Response
Dear Reviewer,
Thank you for your feedback and suggestions, which we have carefully reviewed and taken into account, along with the other reviewer’s comments, during the revision process.
Kindest regards

Round 2
Reviewer 1 Report
Comments and Suggestions for Authors
the author's adressed all my concerns. I have no further suggestions.
Reviewer 3 Report
Comments and Suggestions for Authors
I disagree with the authors when they state that including the information in the introduction ensures accessibility for a broader audience, including readers outside the specific field of study. I believe that high-impact scientific journals are only read by specialists in the field.
I disagree with comment n3. I insist that scientific articles in high-impact journals are only of interest to researchers in the field. Therefore, all content related to "general knowledge" is, in my opinion, irrelevant to include in the article.
I reiterate what I mentioned in my initial review: "In my opinion, the introduction should focus on the content described in the paragraph (lines 83-92) and expand on the scientific evidence (few authors are cited) regarding muscular factors and the subjects' self-perception across different factors."
I stand by my prior recommendation in the earlier review: "I suggest that the authors rewrite the introduction, citing the most important studies on the association between FMS, muscular fitness, self-perception, and physical activity in a structured manner. Then, they should cite the most relevant studies on predictors of FMS competence, athletic competence, and physical activity levels in schoolchildren. After this, I suggest including a paragraph explaining why their study is important." I do not think it is necessary to expand the introduction as the authors propose. Instead, they should simply focus on relevant studies regarding the predictors, an aspect they fail to address.
Participants
In this section, the authors reiterate the objective. Why is this repeated when it is already clearly presented at the end of the introduction?
Thank you for your question. As stated in the manuscript, we have not repeated the objectives. In Section 2.1, we compared the current study to previous studies, as required for this manuscript”.
I insist that the objective is repeated. Even if that were not the case, it makes no sense to compare the current study with previous studies in the description of the sample. This section is not the appropriate place for such comparisons.
BMI Calculation
I completely disagree with the authors regarding BMI calculation. Although the formula is applied, the data must then be adjusted using percentile tables based on the child's age and sex. Therefore, the standard adult BMI formula should not be used without proper adjustments for children.
Statistical Analysis
I insist on this: "In Section 2.3. Statistical Analysis, the authors should describe only the tools used for statistical estimates, without including references to results, such as in lines 250–251: ‘Based on the results of t-tests that revealed significant differences between males and females for six dependent variables (see Table 2).’ This information belongs in the results section and does not fit here.
I disagree with the authors’ response to this recommendation.
Correlation Coefficient
I also disagree with the authors' statement: “Values of r ≥ 0.10 and < 0.30 indicate a small correlation, r ≥ 0.30 and < 0.50 a medium correlation, and r ≥ 0.50 a large correlation.”
A Pearson correlation coefficient of 0.1 indicates that the variables share only 1% of the information; with r=0.3, they share 9%; and with r=0.5, 25%. These are residual proportions, insufficient to establish correlations, especially for predictions. I suggest a minimum of r=0.7 for relevant correlations.
The Pearson correlation coefficient is an objective measure. For example, if it explains 20% of the variance, it is a small correlation. Regardless of how it is analyzed or interpreted, it is an objective value.
Coefficient of Determination (R²) and Table 2
As I mentioned, the R² value in Table 2 explains very little variance, resulting in low predictive power. The authors state: “Although the R² value is an important statistical measure, it is not the sole determinant of a study's relevance. The primary focus is to understand the relationship between FMS and athletic perception, and the moderate R² value reflects that this relationship is part of a broader system. We believe emphasizing the R² value could distract from the broader contributions of the study.”
I disagree with this statement. The R² value (coefficient of determination) is critical because it indicates how much of the model is explained. This is directly related to the Pearson correlation coefficient, which shows the correlation between pairs of variables. If the r values are low, the same will apply to R².
Conclusions
The only actual conclusion provided by the authors is: “Muscular fitness, self-perception, athletic self-concept, and FMS are significantly related variables in primary school students aged 8-10 years.”
The rest of the text in the conclusions is not genuinely conclusions:
"The relationships may be modulated by children's biological sex, suggesting the need for specific strategies in physical education programs. Enhancing our understanding of these relationships can better support children's physical and psychological development, potentially promoting a healthier and more active lifestyle. While further exploration of these relationships among children is warranted, children should have access to PE curriculum opportunities that develop a comprehensive range of FMS and muscular fitness to facilitate confident and competent students, allowing increased participation in diverse physical activities and sports, thus promoting a healthy, active lifestyle."
Summary
Neither the results nor the interpretations of the statistical estimates support these claims. Therefore, I cannot recommend this paper for publication in a high-impact scientific journal.